# Psychogenic Pseudosyncope: Real or Imaginary? Results from a Case-Control Study in Myalgic Encephalomyelitis/Chronic Fatigue Syndrome (ME/CFS) Patients

**DOI:** 10.3390/medicina58010098

**Published:** 2022-01-09

**Authors:** C. (Linda) M. C. van Campen, Frans C. Visser

**Affiliations:** Stichting CardioZorg, Planetenweg 5, 2132 HN Hoofddorp, The Netherlands; fransvisser@stichtingcardiozorg.nl

**Keywords:** chronic fatigue syndrome, myalgic encephalomyelitis, tilt table test, cerebral blood flow, psychogenic pseudo syncope, orthostatic intolerance, syncope, extra cranial Doppler

## Abstract

*Background and objectives*: Orthostatic intolerance (OI) is a clinical condition in which symptoms worsen upon assuming and maintaining upright posture and are ameliorated by recumbency. OI has a high prevalence in patients with myalgic encephalomyelitis/chronic fatigue syndrome (ME/CFS). Exact numbers on syncopal spells especially if they are on a weekly or even daily basis are not described. Although not a frequent phenomenon, this symptomatology is of very high burden to the patient if present. To explore whether patients with very frequent (pre)syncope spells diagnosed elsewhere with conversion or psychogenic pseudosyncope (PPS) might have another explanation of their fainting spells than behavioral psychiatric disorders, we performed a case–control study comparing ME/CFS patients with and without PPS spells. *Methods and results*: We performed a case–control study in 30 ME/CFS patients diagnosed elsewhere with PPS and compared them with 30 control ME/CFS patients without syncopal spells. Cases were gender, age and ME/CFS disease duration matched. Each underwent a tilt test with extracranial Doppler measurements for cerebral blood flow (CBF). ME/CFS cases with PPS had a significant larger CBF reduction at end tilt than controls: 39 (6)% vs. 25 (4)%; (*p* < 0.0001). Cases had more severe disease compared with controls (chi-square *p* < 0.01 and had a *p* = 0.01) for more postural orthostatic tachycardia syndrome in cases compared with controls. P_ET_CO_2_ end-tilt differed also, but the magnitude of difference was smaller than compared with the CBF reduction: there were no differences in heart rate and blood pressure at either end-tilt testing period. Compared with the test with the stockings off, the mean percentage reduction in cardiac output during the test with compression stockings on was lower, 25 (5) mmHg versus 29 (4) mmHg (*p* < 0.005). *Conclusions*: This study demonstrates that in ME/CFS patients suspected of having PPS, or conversion, CBF measurements end-tilt show a large decline compared with a control group of ME/CFS patients. Therefore, hypoperfusion offers an explanation of the orthostatic intolerance and syncopal spells in these patients, where it is clear that origin might not be behavioral or psychogenic, but have a clear somatic pathophysiologic background.

## 1. Introduction

Cardinal symptoms of myalgic encephalomyelitis/chronic fatigue syndrome (ME/CFS) [1,2,3] are fatigue, postexertional malaise, cognitive dysfunction and orthostatic intolerance. Fatigue is the cardinal symptom in CFS, postexertional malaise (PEM) in ME. Cognitive dysfunction is a highly prevalent and classifying symptom in both the CFS and ME criteria [4,5]. Orthostatic intolerance is also highly prevalent, affecting up to 90% of adults and >95% of pediatric patients in the ME/CFS population [6,7]. In the literature, a report of an incidence of 43% of syncope was mentioned in adolescence, which is an overestimated number compared with the incidence in adults [8]. Exact numbers on syncopal spells, especially if they are on a weekly or even daily basis, are not described. Although not a frequent phenomenon, this symptomatology is of a very high burden to the patient if present. This small group of patients usually seeks medical care on a frequent basis and has been seen on more than one occasion by neurologists and cardiologists, and has multiple emergency room presentations. Despite that, the cause of those spells remains unknown, and the diagnosis of behavioral or psychiatric disease is frequent in this patient group [9,10]. As psychogenic syncope may occur in some patients, great care and caution should be made as many of those patients may have an organic cause not identified underlying their problem, as has been shown by heart block/asystole abnormalities on implanted loop recorders [11].

Psychogenic pseudosyncope (PPS) was defined in the 2017 AHA/ACC syncope guidelines as “a syndrome of apparent loss of consciousness (LOC) occurring in the absence of impaired cerebral perfusion or function” [12]. It hasbeen suggested in these guidelines that symptoms patients experience are not voluntary but result from a response to psychological stress. The PPS spells are presumed to be part of a conversion disorder [13].

As described in the IOM report: “Orthostatic intolerance is defined as a clinical condition in which symptoms worsen upon assuming and maintaining upright posture and are ameliorated (although not necessarily abolished) by recumbency” [1]. Symptoms of orthostatic intolerance sought in the history of patients “are those caused primarily by [1] cerebral underperfusion (such as light-headedness, near-syncope or syncope, impaired concentration, headaches, and dimming or blurring of vision), or [2] sympathetic nervous system activation (such as forceful beating of the heart, palpitations, tremulousness, and chest pain. Other common signs and symptoms of orthostatic intolerance are fatigue, a feeling of weakness, intolerance of low-impact exercise, nausea, abdominal pain, facial pallor, nervousness, and shortness of breath” [3]. In a study of 429 adult ME/CFS patients, we demonstrated that, during a 30 min head-up tilt table test, 90% had an abnormal cerebral blood flow (CBF) reduction as assessed by extracranial Doppler measurements [7].

To explore whether patients with very frequent (pre)syncope spells diagnosed in other clinics with conversion or even PPS elsewhere might have another explanation of their fainting spells than behavioral psychiatric disorders, we studied 30 cases of ME/CFS patients with PPS and compared them with 30 gender, age, disease duration matched ME/CFS patients without PPS or (pre)syncopal spells.

## 2. Materials and Methods

### 2.1. Participants

The case–control study was conducted in the outpatient clinic of the Stichting CardioZorg, a cardiology clinic in the Netherlands that specializes in diagnosing and treating adults with ME/CFS. Cases were eligible if they reported having a diagnosis of syncope related to conversion disorder or had a diagnosis of PPS at another center.

The controls were identified from the clinic database of ME/CFS patients who visited our clinic between October 2017 and October 2021, in whom a tilt test was performed for quantification of orthostatic intolerance (OI). ME/CFS controls were matched to the PPS cases first by gender, then by age (±1 years), selecting the closest matching patient to the case. Because our previous research [14] showed that patients with hypermobility have a larger cerebral blood flow reduction during head-up tilt, and with 11 of the PPS cases being diagnosed with hypermobility, we also ensured that 11 control subjects met criteria for joint hypermobility. Patients were considered hypermobile if the diagnosis of joint hypermobility or hypermobile Ehlers-Danlos Syndrome (hEDS) had been made by a geneticist, rheumatologist, or specialized rehabilitation physician. In all other patients seen during the study period in whom a formal diagnosis of hypermobility had not been established, we asked whether they were highly flexible or were hypermobile. In the event of a positive answer, the Beighton score was obtained [15]. For this study, we chose a conservative, elevated Beighton score of 6 or higher as the threshold for confirming the diagnosis of hypermobility [15,16]. The diagnosis of ME/CFS was made according to the ME/CFS criteria of the International Consensus Criteria (ICC) [1,2], and we excluded those with any other illnesses that could explain the symptomatology. From ICC we also scored clinical disease severity; this study excluded the very severely ill disease ME/CFS patients as they were not able to undergo a tilt test [2]. The medical record showed no trauma of the head and neck in the history of patients. Moreover, we scored ME/CFS symptoms based on the IOM criteria [3]. We noted if study participants were using medications that could alter heart rate (HR) or blood pressure (BP); these drugs were discontinued before performing the tilt test.

The study was carried out in accordance with the Declaration of Helsinki. All ME/CFS participants and healthy controls gave informed, written consent. The study was approved by the medical ethics committee of the Slotervaart Hospital, Amsterdam, the Netherlands (P1450).

Tilt testing: hemodynamic measurements of cerebral blood flow, cardiac output and outcome definitions Methodology used has been described elsewhere in related articles. Extended information is available in Appendix A.

### 2.2. Statistical Analysis

Data were analyzed using Graphpad Prism version 6.05 (Graphpad software, La Jolla, CA, USA). All continuous data were tested for normal distribution using the D’Agostino-Pearson omnibus normality test, and presented as mean (SD) or as median with the IQR, where appropriate. Nominal data were compared using the chi-squared test. Between-group comparison was carried out by the unpaired *t*-test. In both groups disease duration was not normally distributed, and between-group comparison was carried out using the Mann–Whitney test. Nominal data as severity and hemodynamic tilt-test results were also compared with the chi-squared test (3 × 2 table). Due to the multiple comparisons, we chose to use a *p* value of <0.01 to be significant.

## 3. Results

During October 2017 and October 2021, 30 cases were diagnosed in their history before presentation with PPS. In our clinic all were diagnosed with ME/CFS and all had orthostatic intolerance with syncopal spells. Those 30 patients were the cases. From our database during the same period, we collected gender, age, disease duration and hEDS matched controls with also confirmed ME/CFS. The matching process was successful. All ME/CFS patients fulfilled the criteria for CFS and the IOM criteria. We will present results on the case and control groups and present one case with example images of supine and tilt-test findings. Of a total of 1200 tilt tests, 30 patients, 2.5% experienced frequent (pre)syncopal spells and were diagnosed in other clinics as having PPS or conversion disorder. All of those had a clinically important abnormal cerebral blood flow reduction at end-tilt (see Table 1 and Figure 1).

Both case and control groups included 28 female and 2 male patients; 11 of those in both groups also received a diagnosis of hEDS. No differences were found in baseline characteristics. The mean age for both groups was 33 (11) years. Disease duration had a median duration of 11 years (IQR 6–21 years). The case group had no one with mild disease according to ICC, 10 with moderate and 20 with severe disease. The control group included 4 with mild disease, 18 with moderate and 8 with severe disease (chi square 11.43, 2 *p* < 0.01). The hemodynamic results of the tilt test of the case group included 7 patients with a normal heart rate and blood pressure response, 2 with delayed orthostatic hypotension and 21 with postural orthostatic tachycardia syndrome (POTS); the control group included 16 patients with a normal heart rate and blood pressure response, 4 with delayed orthostatic hypotension, and P_ET_CO_2_ and cerebral blood flow supine was also found between the groups with respect to cardiac output supine and end tilt.

Table 1 shows the hemodynamic results of the tilt test. No differences were found in heart rate, systolic blood pressure, diastolic blood pressure and cardiac output supine and at end tilt. No differences were also found in supine cerebral blood flow and P_ET_CO_2_. A significant difference was found at end-tilt in both P_ET_CO_2_ as cerebral blood flow. Figure 1 shows graphically the percent reduction in cerebral blood flow in both groups. The case group with PPS had a much larger reduction in cerebral blood flow. The difference in reduction in P_ET_CO_2_ was less compared with the difference in cerebral blood flow, thus not being completely responsible for the cerebral blood flow decline difference between the groups.

### Case Example

A previously healthy female sustained three episodes of closed head trauma at age 16, after which she developed recurrent syncope, daily fatigue, post-exertional malaise and other chronic symptoms. At age 34, a clinician suggested that she satisfied criteria for ME/CFS, with co-morbid fibromyalgia. Investigations directed at the etiology of her syncopal spells were repeated on several occasions over the 33-year period, including tilt table testing with EEG monitoring, but were felt to be normal. She was diagnosed at age 22 with psychogenic pseudosyncope and conversion.

She presented for a second opinion to the Stichting CardioZorg clinic. Her ME/CFS was in the moderate to severe category [2]. During the intake evaluation, which lasted one hour, due to the complexity of her course, she developed a typical presyncopal spell, characterized by lightheadedness but with normal blood pressure and heart rate, without complete loss of consciousness, but having her eyes closed and being less responsive. She improved only after several minutes of lying supine. She then underwent 10 min of head-up tilt table testing with concomitant end-tidal CO_2_ and combined carotid and vertebral artery Doppler measurements of cerebral blood flow, as described elsewhere. The tilt test provoked her usual orthostatic symptoms, along with POTS, hypocapnia during standing, and a severe 40% decline in cerebral blood flow. The same “spell” symptoms commenced during standing with the end-tilt extracranial Doppler measurements being put in place. Figure 2 shows the extracranial Doppler findings supine and end-tilt for the left and right carotid artery supine and end tilt. For her orthostatic intolerance she was treated with pyridostigmine and fludrocortisone, which were associated with a reduction in orthostatic complaints and decrease in the number of (near)-syncope attacks.

## 4. Discussion

The main finding of this case–control study is that ME/CFS patients with frequent disabling (pre)syncopal spells having had a previous diagnosis of conversion disorder or psychogenic pseudosyncope, have a very important cerebral blood flow reduction during tilt-testing when compared with age, gender and disease-duration-matched ME/CFS patients. In contrast to vasovagal syncope (VVS), psychogenic syncope (or psychogenic pseudosyncope, PPS) has a lower prevalence as a cause of loss of apparent loss of consciousness. This explanation of loss of consciousness is also less studied than VVS has been.

Some points of this study need emphasis. First, several reports on PPS suggest that the diagnosis might be insufficiently recognized. Several studies also mention that the definition of PPS is a transient loss of consciousness [12,17,18]. To the best of our knowledge at the centers of the respective authors no studies on cerebral blood flow are carried out. In this case, no evidence will be available documenting cerebral hypoperfusion, so a conclusion of having no somatic pathophysiology as basis of the “syncopal spells” as we and others have suggested might be present, should not be carried out [7,11]. Using video equipment and having EEG measurements present during tilting is not documentation of cerebral hypoperfusion, whereas transcranial Doppler and extracranial Doppler techniques might be able to document cerebral hypoperfusion. A study of Raj et al. showed one example of syncope and one of PPS with transcranial Doppler, where the PPS subject had no abnormalities [19]. We were not able to find a manuscript on a series of PPS subjects being tilted with transcranial Doppler. This study is the first to describe results of cerebral perfusion measurements using extracranial Doppler in a group of ME/CFS patients with the diagnosis of PPS. Kanjwal and colleagues have recommended caution before diagnosing syncopal spells as psychogenic after identifying heart block and prolonged asystole using an implantable loop recorder in three individuals who had been misdiagnosed as psychogenic [11]. Our report suggests that another under-utilized measurement that can help prevent misdiagnosis is the combined carotid and vertebral artery Doppler flow measurement.

Second, a review by Tannemaats and colleagues [20] suggests an incidence of PPS among those with VVS in reported literature ranging from 0 to 12.1%. In our patient population of ME/CFS we found an incidence of 2.5% (30 of 1200 tilt-test). Walsh et al. [13] found an incidence of 14 out of 1401 patients resulting from examining medical records. Follow-up of this patient group showed that half had vasovagal syncopal (VVS) attacks besides the PPS attacks. They suggest that a change in the frequency of attacks and a prolonged duration of the loss of consciousness should alert physicians to this diagnosis. Blad et al. [21] found an incidence of 5% of PPS and 2% were of mixed etiology of VVS and PPS confirming the finding of Walsh et al. that both VVS and PPS “attacks” could be documented in the same patients [13]. The authors suggest that atypical triggers, episodes without prodromes [21,22] and eye closure during episodes of TLOC are more frequently found in PPS. Therefore, incidence varies among study groups, but is low compared with vasovagal syncope and as literature reports. Thus, attacks of VVS and PPS can coincide in the same patient. Our study report on 429 ME/CFS patients shows that differences in hemodynamic outcomes as for instance hypotension or POTS can be present in ME/CFS patients with a high incidence of orthostatic intolerance [7]. Although this study excluded the patients who fainted on the tilt-test, an abnormal cerebral blood flow decline was present even without syncope, and was also present in patients where the tilt test did not show any changes in heart rate or blood pressure, fitting definitions of hypotension or POTS. Furthermore, this manuscript showed, that a high incidence of the “atypical” prodromal symptoms was part of the presentation of orthostatic symptoms (Figure 5) [7]. Therefore, although literature reports that VVS and PPS can coincide in one patient, conclusions on diagnosis cannot be based on having atypical prodromal symptoms, but should include additional investigations as for instance transcranial Doppler or extracranial Doppler measurements. Basing the diagnosis on symptomatology and hemodynamic results on tilt testing has been proven to miss cases.

Thirdly, characteristics and signs can be indicative for the diagnosis of PPS, according to available literature. Heyer et al. report a higher chance of having PPS with a high rate (>20) of prior fainting spells (odds ratio 54.2), ≥2 fainting spells in a single day (odds ratio 29.30 and self-reported loss of consciousness ≥ 2 min (odds ratio 38.8) [22] in a pediatric/adolescent population. This manuscript reports on prodromal symptoms: typical or atypical [23,24]. Prodromal symptom categorized as “typical” for syncope were lightheadedness, vision changes, nausea, and sweating and prodromal symptoms were categorized as “atypical” for syncope included tremor/shaking, difficulty breathing, muscle weakness, headache or pain, stuttering, and the sensation of imminent sleep. The presence of ≥2 atypical prodromal symptoms was considered having an odd ratio for PPS of 6.9 in the studied group [22]. In contrast, a recent study of us showed that recovery of cerebral blood flow after being tilted back to the supine potion can be delayed dependent on the clinical severity classification of the patients [25]. As a higher percentage of the case group (with PPS) had a more severe clinical presentation of ME/CFS, it can be expected that prolonged recovery of symptoms as non/less responsiveness can be due to the fact that hypoperfusion normalizes only slowly in these patients. Therefore, this might indicate that non-responsiveness or too slow recovery of non-responsiveness is not behavioral or psychogenic of origin, but due to the fact that the upright cerebral hypoperfusion recovers slowly in more severely ill ME/CFS patients.

Fourthly, besides symptomatology reported in the literature as typical or atypical prodromal symptoms, a distinction in typical or atypical triggers for syncope has been described [21]. The authors of this manuscript describe classical vasovagal triggers as “pain, emotion, standing up, prolonged standing/sitting, cessation of exercise or recent meals”. Atypical triggers for vasovagal syncope in this manuscript were “exercise, or supine position in the absence of a concomitant trigger such as venipuncture, episodes without prodromes, eye closure during apparent TLOC, shivering or heavy breathing during apparent TLOC, inability to prevent any episode of apparent TLOC by sitting or lying down, and delayed recovery of consciousness. This group defined “a delayed recovery of consciousness if any of the following features was recorded: (1) failure to regain consciousness in the supine position; (2) recovery of consciousness requiring tactile stimulation; (3) attempts by bystanders to resuscitate the patient; or (4) a prolonged period of confusion after the event. As with the atypical prodromal symptoms, in this case for both typical as atypical triggers for syncope, orthostatic stress is mentioned. In the ME/CFS patient population, orthostatic stress gives an abnormal cerebral blood flow reduction during sitting and standing and even with low orthostatic stress abnormalities have been measured and reported [7,26,27]. Before diagnosing a patient with conversion or PPS, proper studies looking into cerebral hypoperfusion are recommended to be performed in these patients. Treatment is completely different and the treatments of PPS may be detrimental for ME/CFS patients.

Finally, some comments on the selected case presentation. The individual patient we report was diagnosed elsewhere as having PPS. She had a wide range of symptoms consistent with ME/CFS, including “atypical” orthostatic intolerance symptoms as defined by Heyer et al. [22]. She had frequent (pre)syncopal spells with increased orthostatic stress and particularly only related to orthostatic stress (prolonged sitting, short standing periods), no other stress factors triggered the spells. No involuntary movements were present, but she was less responsive, which is probably to be expected with a 40% reduction in cerebral blood flow. The center that diagnosed PPS with this patient did this on video observation, the recognition of the “attack” by the patient and her sister and no abnormalities in heart rate and blood pressure during tilt-testing. Furthermore, we speculate that the misperceptions that ME/CFS has a behavioral or psychiatric origin may have contributed to the long delay before the proper diagnosis was reached. Fainting spells and orthostatic intolerance have been recognized as prominent in ME/CFS, and orthostatic intolerance is now considered a core symptom in some ME/CFS illness definitions [3]. Our patient’s syncopal and pre-syncopal spells were present not just with standing during a tilt table test, but also with prolonged periods of sitting. The first spell at the clinic during prolonged sitting, she had her eyes closed and was less responsive. She did not lose consciousness completely. Responsiveness recovered slowly after putting her in a supine position. She had a clinically important 40% reduction in cerebral blood flow while standing during the tilt test, associated with provocation of her typical orthostatic symptoms. The same “spell” symptoms commenced during standing with the end-tilt extracranial Doppler measurements being put in place. We conclude that the fainting spells were not psychogenic in origin, but related to the large decline in cerebral blood flow which we have described could be present standing as well as during sitting [7,26].

### Limitations

We acknowledge that referral bias by the general practitioner may have played a role, selectively referring patients with orthostatic symptoms. Furthermore, as the hypothesis of hypoperfusion as an explanation for PPS, only a small population has been studied yet. Larger patient groups need to be prospectively studied. Furthermore, we did not investigate patients with psychogenic pseudo epileptic seizures. In the context of ME/CFS these patients, also often confronted with a behavioral and conversion diagnosis, might have a similar somatic background with respect to a larger decline in cerebral blood flow during orthostatic stress.

## 5. Conclusions

This study demonstrates that in patients suspected of having PPS, especially those with ME/CFS (patients, who are often being diagnosed with having conversion disorder), cerebral blood flow measurements at end-tilt show a large decline compared with a group of age-, gender- and disease-duration-matched ME/CFS patients. Therefore, cerebra hypoperfusion offers a clear-cut alternative explanation of the orthostatic intolerance complaints and syncopal spells in these patients. The abnormal cerebral blood flow reduction while being upright makes it clear, that the origin of the attacks might not be behavioral or psychogenic, but have a clear somatic pathophysiologic background. This must be taken into consideration in patients with frequent syncopal spells and the extracranial Doppler measurements before and at end-tilt have the potential to improve the evaluation of recurrent syncope and presyncope, and might avoid misdiagnosis of a behavioral origin.

## Figures and Tables

**Figure 1 medicina-58-00098-f001:**
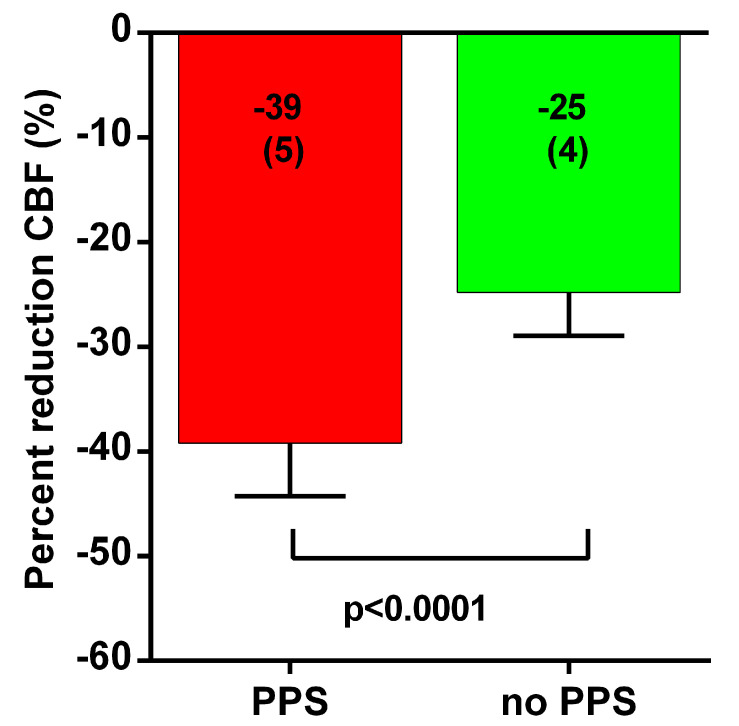
Percent reduction cerebral blood flow end-tilt versus supine in patients with psychogenic pseudosyncope (PPS) (red bar) and patients without psychogenic pseudosyncope (green bar).

**Figure 2 medicina-58-00098-f002:**
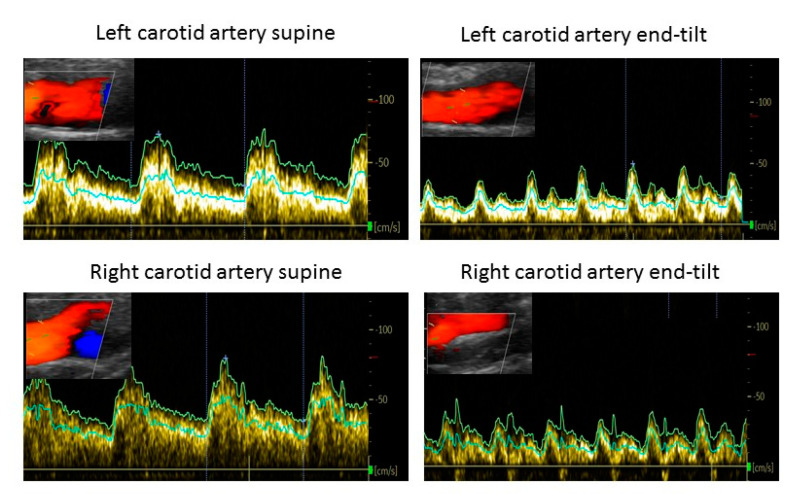
Example of cerebral blood flow measurements of the left carotid artery supine (**upper left**) and end-tilt (**upper right**) and of the right carotid artery supine (**lower left**) and end-tilt (**lower right**) of the example case of: “PPS”.

**Table 1 medicina-58-00098-t001:** Hemodynamic tilt test characteristics.

	Cases PPS	Controls No PPS	*p*-Value
Heart rate supine (bpm)	80 (14)	71 (10)	ns
Heart rate upright (bpm)	111 (22)	101 (21)	ns
SBP supine (mmHg)	134 (17)	133 (11)	ns
SBP upright (mmHg)	126 (20)	127 (20)	ns
DBP supine (mmHg)	81 (13)	80 (11)	ns
DBP upright (mmHg)	87 (14)	85 (14)	ns
CO supine (L/min)	4.96 (0.87)	4.63 (0.79)	ns
CO upright (L/min)	3.68 (1.08)	3.59 (0.67)	ns
CBF supine (mL)	616 (76)	601 (118)	ns
CBF upright (mL)	374 (52)	451 (86)	<0.0001

Footer Table 1; BP: blood pressure; CBF: cerebral blood flow; CO: cardiac output; P_ET_CO_2_: end tidal carbondioxide pressure; PPS: psychogenic pseudosyncope.

## Data Availability

The datasets analyzed in the current study are available from the corresponding author on reasonable request.

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
