# Peer review of "Psychogenic Pseudosyncope: Real or Imaginary? Results from a Case-Control Study in Myalgic Encephalomyelitis/Chronic Fatigue Syndrome (ME/CFS) Patients"

_medicina, 2022, doi:10.3390/medicina58010098_

Round 1

Reviewer 1 Report

The results of this paper show that ME/CFS patients diagnosed with psychogenic pseudosyncope suffer from orthostatic dysregulation manifesting as a severe fall in cerebral blood flow. This misdiagnosis potentially exposing patients to a wrong treatment and social stigmatisation.

A number of corrections are needed:

Line 21: units are not given, or is it %

Line 24: “re”; should be “there”?

Line 44-45: “exact numbers….is not described” plural or singular?

Line 46-47: “this small group….seeks …and have been seen” singular or plural?

Line 50 and line 83: “diagnoses” should be “diagnosis”

Line 53: “there” instead of “their”

Line 64: light- headedness (blank)         

Line 73: “elsewhere” 2x in a sentence

Line 103: “If” capital letter

Line 134: “:2.5%.” could it be placed in parenthesis behind “30 patients”?

Line 151: too many blanks and “..both….as …” instead of “both….and…”. Significan “ for” instead of “in”

Line 166: “which  a she developed”

Line 168: too many blanks

Line 174: “which lasted 1.0 hours”

Line 179: “…Doppler measurement”, should it be plural?

Line 194: “a previous diagnoses”; “a ...diagnosis”

Line 198: “..less studied cause of loss of apparent loss of consciousness” incomprehensible

Line 200:. “First, Several..” Capital letter.

Line 204: “…so no real proof of absence of cerebral perfusion is not present.” The sentence should be rephrased.

Line 205-207: “---the optimal choice of cerebral hypoperfusion..” is it the optimal choice of methods for detecting cerebral hypoperfusion?

Line 208 and 210: “trans cranial” instead of “transcranial”

Line 219: “VVS”, abreviation only explained later in line 222

Line 225: “….Found”. Capital letter; “Blad et al” or “Blad et al.”? several times in the manuscript. Please check.

Line 225-227: “…that both the entitities could be documented in cases”.  Does it mean that “both entities can occur in the same patient”?

Line 230: “…reports, VVS and PPS can coincide..” This is either a new sentence or should be separated by semicolon instead of comma. “The study report on 429 ME/CFS patients…” should it be “Our study report..” or “A study report…”? This sentence is difficult to understand and should be rephrased. Does it mean that a common finding in these ME/CFS patients with ME/CFS and OI is a decrease in CBF although hemodynamic responses to orthostatic challenge can be very different among patients (tachycardia vs hypotension vs. no apparent change in BP/HR)?

Line 234: “Furthermore showed this manuscript..”. “Furthermore, this manuscript showed….”

Line 242: “prodromal symptom….were..”

Line 244: too many blanks

Line 271: “psychogenic pseudosyncope”. Please consistently use abbreviation throughout the manuscript

Line 270-272: both sentences should be rephrased. “patient” and “patients” appear twice in a sentence, uncessarily.

Line 306: see line 194

Line 313: “clear cut” should be “clear-cut”

Line 312-315: this sentence is difficult to understand, and singular and plural are mixed. It should be rephrased. Why is “might” used when your write that “it is clear”? Hypoperfusion offers a clear-cut somatic pathophysiological somatic cause and explanation for OI and syncopal spells excluding a psychogenic or behavioural origin.

The paragraphs starting with “second and tirdly” would benefit from a final sentence or conclusion.  The authors give a lot of detailed information but in the end it is not very clear what they want to say. Is the message of these paragraphs that the presence of atypical prodomal symptoms should not automatically lead to the diagnosis of PPS as atypical prodromal symptoms are part of the symptoms in ME/CFS patients with OI and the documented decrease in CBF by your study? And that the delay in recovery of the fall in perfusion being supine again can influence symptomatoloy and responsiveness that may otherwise be misinterpreted and used as an argument for the diagnosis of PPS?  It would also be useful to have an introductory sentence to each paragraph to already get an idea of the message. 

Author Response

We thank the reviewer for his careful review and suggestions. Reponse is given in the added word document.

Reviewer 2 Report

  • Summary

The authors have carried out a minor-scale case-control study focusing on the cerebral circulation in ME-CFS patients with a diagnosis of psychogenic pseudosyncope (PPS), comparing ME-CFS patients with the aforementioned diagnosis with matched ME-CFS controls not having syncopal spells. The main finding is a significant difference in the reduction of blood flow to the brain between the patient and control groups, which strongly suggests a discord with the guideline definition of PPS; furthermore, the authors are able to point out that the diagnosis of PPS is frequently made on loose ground and in absence of studies measuring perfusion, which is an important and relevant message for the reading public. The authors also describe a case study in detail. The relevance of the discussion is especially clear at present, as the booming ‘functional neurology’ has obvious risks of inaccuracies and even misconceptions of the diagnostic procedures and overall diagnostic thinking. The study has a straightforward design, the study base is quite unequivocally defined, and the study itself is well-performed, clearly reported, and its limitations adequately discussed.  

  • General comments

An obvious limitation is the limited sample, but the matching has been done with care, and notably, the authors have come up with significant differences all the same (even despite some correction for the multiple comparisons). Particularly, the generally high representation of joint hypermobility has been taken into account in both groups, which is commendable. Furthermore, although the phenotyping of syncope is not always simple, by selecting ME-CFS patients with the PPS diagnosis set at other centres, the authors have probably defined an acceptable ‘study base’, and sufficiently precluded patients with non-epileptic seizures or other manifestations.

The main concept of the paper, cerebral blood flow (CBF), is often equated with microcirculation of the brain. Although the measurement of arterial blood flow in all four major precerebral arteries is intuitively a good coverage of the inflow, there is strictly no direct measurement of CBF. The authors have published papers with the same concept previously (and generally, the results have been more consistent than the transcranial flow measurements – which also are surrogates in terms of CBF) but I think this indirect measurement as a surrogate would deserve a comment. An acceptable – and arguably more correct - alternative could be to use the term ‘precerebral blood flow’.

  • Specific comments

Abstract:

  • On rows 14-19 there is redundancy. In the Background and objectives section it would suffice to say that e.g. ‘we performed a case-control study comparing ME/CFS patients with and without PPS spells…´’, and give the details on the rows after Methods and results
  • 21: missing quality of the scalars. P values in parenthesis?
  • 22-23: missing parenthesis

Introduction:

  • 51-53: language needs correction

Methods:

  • 100-1: severity of ME-CFS, how did the groups compare? Later on, it is suggested that the patients had a more severe disease (250-1)

Results:

  • Table 1: CO in the table, CI in the footer
  • Case, 165-66: had the patient craniocervical trauma? The upper spinal (craniospinal) element may have high relevance in this context

Conclusions:

  • 310: ‘accused’ may be a choice with a hint of juridical or at least argumentative undertones...
  • 315-18 might benefit from rewording/linguistic editing

Author Response

(The authors gave the same response as above.)
